# *Salmonella enterica* Outbreaks Linked to the Consumption of Tahini and Tahini-Based Products

**DOI:** 10.3390/microorganisms10112299

**Published:** 2022-11-19

**Authors:** Geneviève Coulombe, Sandeep Tamber

**Affiliations:** 1Microbiology Evaluation Division, Bureau of Microbial Hazards, Food Directorate, Health Canada, 251 Sir Frederick Banting Driveway, Ottawa, ON K1A 0K9, Canada; 2Microbiology Research Division, Bureau of Microbial Hazards, Food Directorate, Health Canada, 251 Sir Frederick Banting Driveway, Ottawa, ON K1A 0K9, Canada

**Keywords:** foodborne illness, foodborne outbreak, low-moisture foods, *Salmonella*, *Salmonella* detection, sesame seeds, tahini, halva

## Abstract

*Salmonella* is a leading cause of bacterial foodborne illness in the world. Although typically associated with foods of animal origin, low-moisture foods, such as tahini, are quickly gaining recognition as an important vehicle of *Salmonella* exposure. This review offers the Canadian perspective on the issue of *Salmonella* in tahini and tahini-based products. A summary of several recent food product recalls and foodborne outbreaks related to the presence of *Salmonella* in tahini and tahini-based products such as halva are presented. The properties of the food vehicles, their production practices, and potential routes of contamination are discussed. Particular focus is placed on the ecology of *Salmonella* in the tahini production continuum, including its survival characteristics and response to intervention technologies.

## 1. Introduction

Salmonellae are an important cause of human illnesses worldwide. It is estimated that nontyphoidal *Salmonella* causes 93.8 million illnesses, of which an estimated 80.3 million are foodborne, and 155,000 deaths each year, worldwide [1]. Most salmonellosis cases are sporadic, meaning they are not linked to known outbreaks [2]. Outbreaks linked to *Salmonella* are often associated with the consumption of foods of animal origin or fresh produce [3]. Multiple outbreaks involving other food commodities have also been reported. Particularly problematic are low-moisture foods, which are defined as foods having a water activity (a_w_) of 0.85 or below. This category of foods includes chocolate, peanut butter, and tahini, all of which have been linked to multiple large outbreaks of salmonellosis across the globe.

Low-moisture foods are generally regarded as low-risk foods because they cannot support the growth of pathogenic bacteria, including those belonging to the genus *Salmonella. Salmonella*, however, has the capacity to survive in low-moisture foods for extended periods, up to several years. This prolonged survival increases the risk associated with the consumption of low-moisture foods. As shelf-stable products, a contaminated batch of low-moisture food has the potential to be ingested over time on multiple occasions by multiple people, leading to a wider temporal and geographic distribution of the pathogen than with other food commodities, and a consequent higher number of cases. Three of the largest salmonellosis outbreaks reported in the published literature are linked to low-moisture foods; dried cuttlefish snacks, paprika, and peanut butter with 1505, 1000, and 715 reported cases, respectively [4,5,6].

In 2004, the World Health Organization’s International Food Safety Authorities Network (INFOSAN) identified sesame-based foods (i.e., tahini and halva from the Middle Eastern region) as an unusual food source for *Salmonella* contamination [7]. To date, twenty salmonellosis outbreaks linked to the consumption of tahini and tahini-based products (e.g., halva and hummus) have been identified worldwide (Table 1) [8]. Overall, these outbreaks resulted in 1662 salmonellosis cases, 88 hospitalisations, and 1 death.

In Canada, from 2011–2021, tahini and tahini-based products (e.g., halva and hummus) have been subject to multiple recalls and updated recalls due to the presence of *Salmonella* [9]. In Canada, salmonellosis illnesses linked to the consumption of tahini imported from Israel and hummus were identified in 2018 and 2020, respectively (Table 1) [10]. In 2019, eight Canadian cases were identified by whole genome sequencing (WGS) as being part of a multicountry outbreak linked to tahini and halva imported from Syria (Table 1). Indeed, tahini, halva, and other products made from sesame seeds were labelled, by the Public Health Agency of Canada (PHAC), as being in the “high-risk” food category due to their potential of being contaminated by *Salmonella* [11].

This review offers the Canadian perspective on the issue of *Salmonella* in tahini and tahini-based products and provides an overview of the available literature concerning the ecology of *Salmonella* in sesame seeds, tahini, and halva. Intrinsic properties of sesame seeds, tahini, and halva will be summarized, along with extrinsic factors that can lead to contamination along the processing continuum. Features of the pathogen allowing for long-range survival, mitigation strategies, and areas for further research will also be covered.

**Table 1 microorganisms-10-02299-t001:** Known salmonellosis outbreaks linked to tahini and tahini-based products, 1995–2022.

	Year	Products	*Salmonella* Serovar	Country	Cases (Hospital/ Deaths)	Country of Origin	References
1	1995	Tahini	Brandenburg	USA	137	NA	[12]
2	2001	Halva	Typhimurium DT104	Australia, New Zealand, Sweden, Norway	62	Turkey	[13,14,15,16,17]
3	2002	Tahini	Montevideo	Australia	55	Egypt	[18]
4	2003	Tahini	Montevideo	New Zealand	10	Lebanon	[18]
5	2003	Tahini	Montevideo	Australia	3	Lebanon	[18,19]
6	2004	Hummus, eggplant dip	Typhimurium DT197	Australia	173 (25)	NA	[8]
7	2007	Hummus	Heidelberg	USA	802 (29)	NA	[20]
8	2007	Hummus	Heidelberg	USA	11	NA	[8]
9	2007	Hummus	Muenchen	UK	6	NA	[21]
10	2010	Hummus	Typhimurium	Australia	45 (8)	NA	[22]
11	2011	Tahini, hummus	Bovismorbificans	USA	23	Lebanon	[23,24]
12	2012	Tahini	Montevideo, Mbandaka, Maasticht	New Zealand	16 (3)	Turkey	[25]
13	2013	Tahini	Montevideo, Mbandaka	USA	16 (1/1)	Turkey	[24,26]
14	2016–2017	Tahini, sesame seeds	Vari (11:z41:e,n,z15)	Germany, Czech Republic, Luxembourg, UK, Greece	47 (12)	Greece	[27]
15	2017	Hummus	Thompson	USA	13	NA	[24]
16	2018	Tahini, hummus	Concord	Israel, USA	45	Israel	[28]
17	2019	Tahini	Concord	USA	6 (1)	Israel	[29]
18	2019	Hummus	*Salmonella*	USA	9	NA	[30]
19	2020	Hummus	*Salmonella*	Canada	45 (9)	NA	[31]
20	2019–2022	Tahini, halva	Mbandaka, Havana, Amsterdam, Orion, Kintambo, Senftenberg	Germany, Sweden, Norway, Netherlands, Canada, USA, New Zealand	138 (24)	Syria	[32,33]

## 2. Salmonella

The genus *Salmonella* consists of two species, *S*. *enterica* and *S*. *bongori*, that have been subtyped into over 1500 serological variants, or serovars. In terms of human illness, multiple serovars of *S*. *enterica* account for over 99% of human salmonellosis cases. Children under 5 years old, the elderly, and people with compromised immunity are most vulnerable to salmonellosis and more likely to develop severe symptoms [34,35].

Estimates of the infectious dose for *Salmonella* range from 10^0^ to 10^11^ cells [36,37]. Host-related factors such as age and immune status account for much of this variability. The virulence properties of individual strains and the composition of the food matrix can also influence the number of bacteria required to produce an infection. There is evidence that food matrices with a combination of high fat and low water activity (a_w_) in a food matrix may protect *Salmonella* from the acidic conditions of the stomach, thus increasing the likelihood of illness from consuming low numbers of the microorganism [38,39]. Analyses of outbreak-associated, high-fat, low-moisture foods such as chocolate and potato chips indicate that inocula ranging from 1 to 45 cells can lead to symptomatic infections [4,40,41,42].

Concentrations of *Salmonella* ranging from <0.03 MPN/g to 0.46 MPN/g have been found in tahini and tahini-based products linked to salmonellosis outbreaks [18,25]. Based on these levels and the infectious doses reported above, the consumption of as little as 2 g of contaminated product has the potential to lead to illness.

## 3. Outbreaks Due to the Presence of *Salmonella* in Tahini and Tahini-Based Products

The first reported salmonellosis outbreak linked to the consumption of tahini occurred in 1995 in the United States. A total of 137 individuals got sick due to the presence of *Salmonella* ser. Brandenburg (Outbreak #1, Table 1). Following that, in 2001, there was an international outbreak of *Salmonella* ser. Typhimurium DT104 due to consumption of halva, a ready-to-eat confectionary made with tahini. Cases were reported in Europe as well as Australia and New Zealand (Outbreak #2, Table 1). In 2002 and 2003, three outbreaks of *Salmonella* ser. Montevideo linked to the consumption of tahini occurred in Australia and New Zealand. The implicated tahini products were manufactured in Egypt and Lebanon (Outbreaks #3 to 5, Table 1). Of the twenty outbreaks listed in Table 1, country-of-origin information is available for eleven. These countries include Egypt, Greece, Israel, Lebanon, Syria, and Turkey as the source of the tahini and tahini-based products. The source of the sesame seeds, in most cases, remains unknown.

Diverse *Salmonella* serovars have been isolated during the course of these outbreak investigations, including some reported in multiple outbreaks: Concord (two outbreaks), Heidelberg (two outbreaks), Mbandaka (three outbreaks), Montevideo (five outbreaks), and Typhimurium (three outbreaks). In 2016 and 2017, an outbreak of a previously undescribed *Salmonella* serovar, Vari, occurred in five European Union countries (Outbreak #14, Table 1). The investigation into this outbreak indicated that cross-contamination likely occurred at the Greek manufacturing facility during the production of tahini [27].

Some outbreaks have involved multiple serovars. Six serovars were isolated during the investigation of a multicountry outbreak linked to tahini and halva imported from Syria (Outbreak #20 in Table 1). In that outbreak, cases were identified intermittently from 2019 to 2022 in five European countries (i.e., Denmark, Germany, the Netherlands, Norway, and Sweden), as well as Canada (eight cases), the United States (six cases), and New Zealand (three cases). This outbreak illustrates the worldwide distribution of tahini and tahini-based products. The investigation into this outbreak was not able to determine the root cause of *Salmonella* contamination. However, since the implicated products were in sealed packages, it is likely the contamination event occurred prior to packaging and, hence, exportation [32].

In two instances, illnesses linked to tahini and hummus have occurred in Canada. In 2018, Canadian cases were reported, and the source was identified as imported tahini from Israel. This resulted in a recall [10]. In 2020, a localized outbreak linked to hummus involved a restaurant and a food truck from a single region (Outbreak #19, Table 1). There were 45 laboratory-confirmed cases, and 185 cases which were symptomatic but were not laboratory confirmed. This outbreak illustrates the risks associated not only with tahini, but also with foods prepared from it, such as hummus.

## 4. The Ecology of *Salmonella* in Tahini and Tahini-Based Products

### 4.1. Salmonella and Sesame Seeds

Sesame seeds are cultivated from the sesame plant (*Sesamum indicum*). They are used as cooking and baking ingredients. Their high oil content (approximately 50% of the seed’s weight) makes them a valuable source of oil for cooking, cosmetic, and pharmaceutical applications [13]. Over 95% of the world’s sesame seed crop is produced in Africa (43%) and Asia (53%), with worldwide production estimated at seven million metric tonnes. Sudan, Myanmar, the United Republic of Tanzania, and India were the largest producers in 2020 [43]. Many countries, including Canada, do not produce sesame seeds and, therefore, rely on imported sesame seeds to produce tahini and tahini-based products [43]. Approximately 70–75% of the sesame seeds used in the United States are imported, with a few southern states producing seeds domestically [44,45].

*S*. *indicum* is a flowering annual that grows in areas with an annual rainfall of 625–1100 mm and temperatures of 27 °C and higher [46]. The sesame plant has an extensive root system and favours well-drained fertile soils with a neutral pH. The fruit of a sesame plant is a capsule that contains multiple seeds. Once the seeds are ripe, the capsule splits open and the seeds are released. These shattering cultivars of sesame seeds are largely harvested by hand. Non-shattering variants have been developed and are more amenable to mechanical harvesting [45]. Excess water, rain, and wind can promote shattering and decrease the yield of the seeds.

Contamination of the sesame seed plant can occur at the preharvest stage. *Salmonella* may be present in the soil, irrigation water, or in fertilizer. Droppings from wild animals carrying *Salmonella* are another potential source of the pathogen [47]. After harvest, sesame seeds are dried to an a_w_ of 0.5 [48]. Drying the seeds is a critical step in the production process. It is carried out in open areas that can be exposed to dust and aerosols [49,50]. Wet conditions can complicate the drying process as the size and shape of seed prevents aeration and may prolong the time the seeds are exposed to atmospheric elements.

Many *Salmonella* serovars have been recovered from sesame seeds and include Typhimurium DT104, Offa, Tennessee, Poona [13], Montevideo, Stanleyville, Tilene [51], Amsterdam, Anatum, Bareilly, Charity, Cubana, Gaminara, Tennessee, Hvittingfoss, Kentucky, *S*. *enterica* ssp. *diarizonae* [52], Weltevreden, Newport, Mbandaka, Anatum, Senftenberg, Give, Tennessee, 3, 10: b:-, Havana, Kentucky, Bonn, Cerro, Glostrup, Idikan, Llandoff, Pottsdam, Westminister, *S*. *enterica* ssp. *arizonae*, and *S*. *bongori* 48:z4,z24:- [53]. This diversity suggests multiple sources of contamination. Some of these serovars are known clinically and have been implicated in previous outbreaks linked to low-moisture foods (e.g., Montevideo, Tennessee, Newport, and Poona) [6,54,55,56], whereas others are rarely seen in clinical settings and their virulence properties are unknown (e.g., Tilene, Charity, Idikan, and Amsterdam). It is also not known whether these serovars have geographic significance or unique physiologies that allow them to survive in the sesame seed production environment.

The presence of *Salmonella* in sesame seeds varies widely (Table 2). Surveys of sesame seeds from a variety of locations have shown *Salmonella* prevalence ranging from none detected in 526 samples to 27% out of 359 samples using an analytical unit of 25 g [49]. Studies with enumeration data indicated low levels of contamination with concentrations ranging from 0.06 to 4 MPN/100 g [44] and 360 MPN/100 g [48]. No relationship was observed between *Salmonella* presence and counts of total aerobic bacteria, coliforms, and/or *E*. *coli* [48,52,57,58]. Surveillance studies of sesame seeds for import into the United States over ten years showed a relatively high prevalence in imported seeds (8–11%) compared to seeds that were collected from domestic retail establishments in the USA (no detections). This shows that postprocessing interventions may be able to control the levels of *Salmonella* on sesame seeds [44,52,53]. These studies also demonstrated a high degree of batch-to-batch variation with respect to *Salmonella* presence, indicating the pathogen is not distributed homogenously within the product, and may require larger sample sizes for reliable detection.

### 4.2. Salmonella and Tahini

Tahini is the paste produced from ground sesame seeds, which has a high-fat (57–65%) and low-moisture content (<1%) [69]. On average, tahini has a water activity (a_w_) of 0.16 and a pH of 5.9 [70,71]. Tahini is considered a ready-to-eat product, which is stored at ambient temperature with a long shelf life (up to two years) [18]. Traditional to Middle Eastern and Mediterranean cuisine, tahini is used as an ingredient in the preparation of many other ready-to-eat products, such as hummus, baba ghanoush, mutabbal, tarator sauce, and various salad dressings, sauces, and dips [50,72]. These foods pose an additional risk for acquiring salmonellosis since their high-moisture content could amplify *Salmonella* levels should it be present in any of the raw ingredients.

Countries that are key global exporters of tahini include Lebanon, Syria, Egypt, Greece, and Israel [73]. Tahini is gaining popularity in North American and European markets [74]. As an example, 6.8% of the respondents who participated in Foodbook, a Canadian population-based telephone survey, reported the consumption of tahini, halva, and other products made from sesame seeds in the last 7 days [75]. However, this may be an underestimation since multi-ingredient foods, such as sandwiches and salads, may contain tahini as an ingredient that consumers are not aware of [11].

Tahini is typically obtained by milling cleaned, dehulled, and roasted sesame seeds [50,70]. The process includes multiple steps: an initial soak step, a dehulling step, a draining and drying step, a thermal treatment step (roasting), followed by grinding [76]. Some manufacturers might pasteurize the finished product, although it is not known how widespread this practice is and what parameters are being used [61,77]. Other variations in the process have been reported [13,61]. The soaking step is of critical importance for process control. It can be carried out in water or salt water for 12 to 24 h. If *Salmonella* is present on the initial lot of seeds, this practice can amplify the level of *Salmonella* by up to 3 logs prior to the roasting and grinding of seeds [76].

The majority of tahini sold for consumption is made from roasted seeds. Raw tahini, made from unroasted or lightly roasted sesame seeds, represents a small percentage of all tahini sold (e.g., 4.1% in Canada) [50,62]. Roasting parameters are variable and expected to differ among producers. Different roasting temperatures and times (110 to 170 °C for 40 to 180 min), as well as heat-treatment processes (steam and dry heat roasting treatments), have been reported [78,79,80].

Studies have demonstrated that the roasting of sesame seeds can reduce *Salmonella* levels [76,80]. Torlak and colleagues reported that roasting temperatures and times of 100 °C for 60 min, 130 °C for 50 min, and 150 °C for 30 min were sufficient to generate a minimum of a 5-log reduction in the initial levels of *Salmonella* artificially inoculated onto sesame seeds. The authors noted a period of rapid decline within the first 10 min of roasting, followed by a lower rate of reduction. This first ten minutes of heating corresponded to a reduction in the seeds’ a_w_, from 0.98 to 0.14. This suggests that during the roasting process, the surviving population of *Salmonella* cells exhibit an increase in heat resistance as the moisture of the seeds decreases [80].

Similarly, Zhang and colleagues inoculated sesame seeds with 8.5 log CFU/g of *Salmonella*, then soaked and dried them to an a_w_ of 0.90. When these seeds were roasted at 130 °C using an air-forced oven, *Salmonella* populations decreased below the detection limit (1.7 log CFU/g) within 10 min. However, when the sesame seeds had an a_w_ of 0.45 before roasting at the same temperature, the decline in cell populations took longer with an approximate 5-log reduction after 60 min, meaning that 3.5 log CFU/g remained on the seeds after this roasting process [76]. This article indicates that the lethality of the roasting process to *Salmonella* present on the sesame seeds depends on the starting a_w_.

If *Salmonella* survives the roasting process, it is likely to remain viable throughout the shelf life of the product. Therefore, tahini and tahini-based products may become contaminated by *Salmonella* through insufficient or inadequate roasting of contaminated sesame seeds [76,80]. In addition, it is crucial to avoid cross-contamination after the roasting process. *Salmonella* cross-contamination in low-moisture foods has been traced to factors such as poor sanitation practices, poor equipment design, improper maintenance, and poor ingredient control [47,81].

Guidance documents for manufacturers of tahini and tahini-based products describing *Salmonella* control practices are available. These documents include the Code of Hygienic Practice for Low-Moisture Foods, CXC 75-2015 [39]; the *Hazard Analysis and Critical Control Point Generic Models for Some Traditional Foods: a Manual for the Eastern Mediterranean Region* [70]; and the Control of *Salmonella* in Low-Moisture Foods [81]. The Grocery Manufacturers Association (GMA) of the United States listed the following seven control elements against *Salmonella* contamination: prevent ingress or spread of *Salmonella* in the processing facility, enhance the stringency of hygiene practices and controls in the Primary *Salmonella* Control Area, apply hygienic design principles to building and equipment design, prevent or minimize growth of *Salmonella* within the facility, establish a raw materials/ingredients control program, validate control measures to inactivate *Salmonella*, and establish procedures for verification of *Salmonella* controls and corrective actions [81].

Published *Salmonella* prevalence levels in tahini from Canada, Germany, and the Middle East ranged from 0.4% to 20% (Table 2) [13,60,61]. As with sesame seeds, no relationship was observed between *Salmonella* presence and total aerobic mesophile/coliform/*E*. *coli* counts. A targeted survey performed in Canada from 2010 to 2014 found a *Salmonella* prevalence of 0.4% (out of 2,315 samples) in tahini. All positive samples were from tahini imported from Middle Eastern countries (i.e., Lebanon, Syria, and Israel), even though almost half of the products tested were domestically produced [62,63,64]. It is not clear whether the difference between imported and domestic products was due to the origin of the raw ingredients, or to differences in production practices. A study by Alaouie and colleagues comparing tahini made from traditional or more automated methods did not find a difference in *Salmonella* prevalence among the sampled products [61]. Serovars found in tahini include Amsterdam, Havana, Montevideo, Senftenberg [62,63,64], Typhimurium DT 104 [13], Hadar, Agona, Einsbuettel, and Ubrecht [60].

### 4.3. Salmonella and Halva

Halva is a confectionary widely consumed in the Middle East and Mediterranean. It is primarily a mixture of tahini and sugar with the following specifications: ≥24% fat, ≥8.5% protein, ≤55% sucrose, ≤2% fibre, and ≤3% water [68]. An analysis of halva produced in a Greek facility indicated an a_w_ of 0.18 and pH of 6 [68]. Halva is made by mixing tahini with a heated, acidified sugar syrup. The syrup contains a high concentration of glucose, citric or tartaric acid, and soapwort root extract (*Saponaria officinalis*). The syrup is heated to 120–140 °C prior to mixing with the tahini [13,68]. Nuts, cocoa, and other flavourings can be added before portioning and packaging [65]. These additives, particularly cocoa and pistachio, can act as vehicles for the introduction of *Salmonella* into the product. As an illustration, halva products containing cocoa and/or pistachio demonstrated increased *Enterobacteriaceae*/coliform/*E*. *coli* counts compared to plain halva [65,67]. There was no relationship between the levels of *Enterobacteriaceae*/coliform/*E*. *coli* and *Salmonella* presence, suggesting these counts can be used as indicators of overall process hygiene as opposed to indicators of pathogen presence. Reported *Salmonella* prevalences in halva ranged from no detections to 11.3%, with Typhimurium DT 104, Poona, and a monophasic strain from serogroup B isolated (Table 2) [13].

## 5. Survival of *Salmonella* in Sesame Seeds, Tahini, and Halva

### 5.1. Survival Studies

Water activity (a_w_) is a key determinant for microbial growth as it is a measure of the amount of water available to carry out essential metabolic functions. *Salmonella* requires a minimal a_w_ of 0.93 to actively grow and reproduce [36]. Therefore, it cannot grow in sesame seeds, tahini, or halva, but it can survive in these matrices for extended periods [47,82]. Studies on the survival of *Salmonella* in sesame seeds, tahini, and halva have shown survival periods ranging from at least 4 to 12 months [48,68,76,80,83,84]. These studies highlight the public health risk associated with the presence of *Salmonella* in these foods. Even after prolonged storage, the pathogen remains viable, and therefore, capable of causing disease.

The manner in which tahini and halva are stored appear to have an effect on *Salmonella* survival. Increased survival was noted when tahini and halva were stored at refrigeration temperatures as opposed to room temperature [80,84]. It has been proposed that the increased survival at lower temperatures may be linked to changes in the composition of the bacterial membrane that would result in lower rates of permeability [83]. It is also possible that the shift in temperature changes the organization of the lipid–water interface of the tahini/halva in a manner that supports bacterial survival. Work carried out with peanut butter and margarine has shown that the size of the lipid–water droplets can influence microbial survival, with finer droplets presenting more of a hurdle [85,86]. Whether this relationship applies to tahini and halva remains to be investigated.

Other factors that can affect survival include additives such as acids or essential oils derived from spices, both of which led to marginal reductions in *Salmonella* in tahini stored at 10, 25, or 37 °C for 1 month [72,87]. Product packaging can also affect survival, as higher levels of *Salmonella* were recovered from vacuum-sealed halva after eight months of storage compared to air-sealed packages [68].

### 5.2. Survival Strategies

*Salmonella* employs a number of adaptations to survive in low-moisture environments [47,82]. Upon inoculation in peanut oil, cells enter a partially dormant state where less than 5% of their genome is transcribed (compared to 78% of cellular genomes cultured in Luria broth) [88]. Among the genes that are transcribed in peanut oil are stress response genes as well as the global regulators sigma D and sigma E, suggesting that in this state, persistence and survival take precedence over growth and metabolism. Upon prolonged storage, *Salmonella* cells can become metabolically dormant where they are viable but not culturable (VBNC) until exposed to favourable growth conditions, at which point they can resuscitate and resume metabolic and other accessory functions such as pathogenesis.

Changes to the cell structure and cell surface have been reported in low-a_w_ conditions. *Salmonella* cells grown in high NaCl, sucrose, or glycerol form long filaments due to an inhibition of cell septation. Similarly, high osmolarity conditions lead to an alteration of the OmpC/OmpF porin ratios in the outer membrane that lead to altered permeability rates. Permeability can also be affected through increased synthesis of unsaturated fatty acids via FabA [89]. Osmotic pressure is maintained through the upregulation of genes encoding osmoprotectant synthesis and transport proteins (ProP, ProU, and OsmU), as well as increased potassium influx through the Kdp transporter [88]. However, it is not clear if these adaptations extend to sesame seeds, tahini, and halva since the method of desiccation can influence the nature of the bacterial response [82].

*Salmonella* cells that are tolerant to desiccation undergo a number of physiological changes that may render them less susceptible to other food control measures such as heating, acidification, and/or treatment with biocides [47]. For example, *Salmonella* cells inoculated onto sesame seeds at a starting a_w_ of 0.45 exhibited a slower rate of decline compared to cells on seeds with an a_w_ ≥ 0.95 when the seeds were roasted at 130 °C [76].

There may be a relationship between biofilm formation and desiccation resistance, since cells that form biofilms have increased resistance to many ecological stresses [90]. Cells that overproduce the biofilm extracellular matrix components curli and cellulose have an increased tolerance to desiccation stress [91]. Similarly, mutants deficient in the production of an extracellular matrix have reduced desiccation tolerance [92]. However, it is not clear if this relationship holds in high-fat, low-a_w_ food systems and is an area that requires further study.

The ability of *Salmonella* to tolerate desiccation varies from strain to strain. Norberto and colleagues reported in a study of 37 *Salmonella* strains belonging to 16 serovars log reductions ranging from 0.6 to 2.7 when inoculated in dry soybean meal and stored at 25 °C for 18 h [93]. There were considerable differences within strains of the same serovar; for example, the respective range of log reductions was 0.9, 1.1, and 1.3 for *S*. *enterica* serovars Ohio, Montevideo, and Havana, respectively [93]. The nature of these differences is not clear, but may reflect differences in gene presence or gene expression that can impact the response to desiccation.

## 6. Control of *Salmonella* in Sesame Seeds, Tahini, and Tahini-Based Products

Both thermal and nonthermal treatments of low-moisture foods or their raw materials can be used to control the presence of *Salmonella*. In recent years, there has been intensive research efforts into postproduction lethality treatments of tahini and tahini-based products. The intent is to eliminate *Salmonella* without compromising the quality properties of the food products. Some of the inactivation treatments tested on sesame seeds, tahini, or halva are presented below.

### 6.1. Thermal Treatment

Roasting of the sesame seeds is a key step to control *Salmonella* during the manufacturing of tahini, as described in detail in Section 4.2.

Some manufacturers may also include a thermal treatment applied to the tahini as a final step [77]. Szpinak and colleagues evaluated the efficacy of a final heat treatment by testing the survival of *Salmonella* ser. Typhimurium in tahini treated at temperatures of 70 °C, 80 °C, and 90 °C. Regardless of the test conditions, the *Salmonella* cells had a high rate of survival with a maximal reduction of 3 log CFU/mL after one hour at 90 °C. The inactivation profiles demonstrated a rapid decline in the first two minutes of treatment, followed by a tailing curve at all tested temperatures, suggesting the applied thermal treatments were only able to inactivate a subset of the *Salmonella* population [77].

### 6.2. Natural Antimicrobials

Some studies evaluated the impact of incorporating natural antimicrobials into tahini in order to control *Salmonella* [72,87,94]. Ten plant essential oils were tested for their antimicrobial activity against four different *Salmonella* serotypes (Typhimurium, Aberdeen, Cubana, and Paratyphi A) using a disc diffusion assay method [87]. In that study, thyme oil and cinnamon oil showed the highest antimicrobial activity. When added to tahini, 2.0% cinnamon oil reduced the numbers of *Salmonella* between 2.4 and 2.8 log CFU/mL, whereas 2.0% thyme oil led to a reduction between 2.2 and 3.3 log CFU/mL by 28 days, depending on the storage temperature (10 °C, 25 °C, or 37 °C). However, the addition of essential oil extracts reduced the quality and consumer acceptability of the tahini, as assessed by a sensory panel [87]. Another study investigated the use of 3% oregano oil in tahini, which led to a reduction of 1.4 and 0.80 log CFU/g after 7 days at 25 °C and 4 °C, respectively, compared to the control [94].

The antibacterial effects of organic acids, such as acetic and citric acids, have also been tested in tahini. The addition of acetic and citric acids at a concentration of 0.5% reduced *Salmonella* ser. Typhimurium by 2.7–4.8 log CFU/mL and 2.5–3.8 log CFU/mL, respectively, in tahini after 28 days at tested temperatures of 10 °C, 21 °C, and 37 °C [72]. Similarly, Xu and colleagues confirmed a reduction in *Salmonella* levels with the use of 0.5% citric acid (0.8 log CFU/g reduction after 7 days at 4 °C) [94].

Antimicrobial substances, such as metals, chemicals, essential oils, enzymes, and bacteriocins, can also be used in packaging materials [95]. More research is needed to evaluate their potential effect on *Salmonella* spp. in tahini and tahini-based products.

### 6.3. Irradiation

Ionizing radiation can effectively eliminate microbes that cause foodborne illness, including *Salmonella*. Three types of radiation can be used on bulk or packaged products: gamma rays, X-rays, and electron beams [96]. D’Oca and colleagues evaluated the inactivation of *Salmonella* ser. Montevideo by gamma irradiation of sesame seeds. Their results showed that the use of 5 kGy was able to reduce *Salmonella* counts by 5.53 log CFU/g on sesame seeds [51]. The inactivation of *Salmonella* using gamma irradiation was also tested in tahini [97] and halva [98]. In halva, an irradiation dose of 4.0 kGy resulted in a reduction of 2.1 log CFU/g of *Salmonella*, whereas in tahini, an irradiation dose of 2.0 kGy led to a reduction of 4.7 log CFU/g of *Salmonella* [97,98]. In tahini, the irradiation dose of 2.0 kGy did not affect the physical and chemical properties of the product (i.e., color, peroxide, p-anisidine, and acid values) [97].

### 6.4. Energy-Based Technologies

Two recent studies tested energy-based technologies on sesame seeds and tahini [99,100]. Xu and colleagues tested the use of radiofrequency heating, in combination with cinnamon oil vapor, on sesame seeds. A radio frequency heating at 80 °C for 5 min combined with 0.83 μL/mL of cinnamon oil vapor achieved a reduction of greater than 5 log CFU/g of *Salmonella* ser. Montevideo on sesame seeds. No synergistic effect of the combined treatments was observed, nor was there any reduction in the quality of the seeds [100].

The efficiency of 2450 MHz microwave heating at 220, 330, 440, 550, and 660 W for the inactivation of *Salmonella* was assessed in tahini. *Salmonella* reductions ranging from 3.6 to 4.7 log CFU/g were observed after treatment. After two minutes of treatment, the core temperatures of the tahini samples reached, 45 °C, 70 °C, 103 °C, 119 °C, and 144 °C at 220, 330, 440, 550, and 660 W, respectively. The authors noted that microwave heating did not affect acid, peroxide, p-anisidine, or color values of the tahini up to 90 °C [99].

### 6.5. Fumigation and Gas Treatment

Fumigants, such as ethylene oxide and propylene oxide, have shown to be effective for achieving significant reductions in microbial populations in low-moisture foods. More precisely, ethylene oxide is used to treat spices, whereas propylene oxide is used with a variety of foods such as nuts, spices, cocoa beans, and dried fruits [101].

In some countries, including the European Union, the use of ethylene oxide in foods is not permitted. In recent years, multiple recalls of sesame seed products due to the presence of ethylene oxide residues occurred in the European Union. It is believed that ethylene oxide was used for fumigating sesame seeds to eradicate *Salmonella* contamination [102].

Golden and colleagues tested the use of chlorine dioxide (ClO_2_) to reduce *Salmonella* levels on sesame seeds. A treatment with 500 mg ClO_2_/kg reduced the number of *Salmonella* from 7.6 log CFU/g to 4.9 log CFU/g on sesame seeds [103]. 

## 7. Monitoring

There are many challenges associated with the detection of *Salmonella* in sesame seeds, tahini, and tahini-based products. The low numbers of pathogen and heterogeneous distribution patterns necessitates large sample sizes, often beyond the capacity of many food-testing laboratories [104]. Further, the physiological state of *Salmonella* in the low-moisture food matrix requires consideration to prevent false-negative results. As an adaptation to the low-moisture environment, *Salmonella* cells may become injured, enter a period of dormancy (such as a viable but not culturable state), or adopt a filamentous shape [105,106]. Primary enrichment in a nonselective broth encourages cells to recover from these states and multiply to detectable levels. Approaches such as the addition of protectants or applying slower rates of rehydration to the primary enrichment may be employed to prevent cytolysis and increase the level of viable *Salmonella* [107,108]. Rapid methods that forgo secondary enrichment and/or plating in favour of molecular methods to detect *Salmonella* such as PCR should be used in conjunction with a primary enrichment step [104]. The requirement for primary enrichment often delays the time to result, and further research is required to understand how the enrichment period can be optimized for low-moisture foods. Given the issues with product testing, process and environmental monitoring are essential to ensure the hygiene of the tahini manufacturing process [109].

## 8. Areas for Further Research

In the course of preparing this review, the following knowledge gaps concerning *Salmonella* and tahini were identified. Research directed in these areas can provide information required to mitigate the public health risks associated with the presence of *Salmonella* in tahini and tahini-based products:Understanding the ecology and potential virulence of *Salmonella* serovars associated with sesame seeds, tahini, and tahini-based products;Knowledge of the response of sesame and tahini-associated serovars, and whether findings from different serovars can be generalized to the genus;Survival of *Salmonella* in tahini and other colloidal foods, particularly the impact of rheological and physical properties (i.e., particle size);*Salmonella* adaptations to high-fat, low-moisture foods such as tahini (e.g., biofilms, filaments, viable but not culturable) and the potential impact on detection and lethality treatments;The impact of interventions, alone and in combination, throughout the tahini manufacturing process, particularly at the seed soaking and roasting steps;*Salmonella* response in commercial settings to typical sesame seed roasting parameters, in order to understand the physiology of cells that survive the roasting process;Improved traceability and environmental/process monitoring of sesame seeds, tahini, and tahini-based products to allow for a more detailed root cause analysis of food safety incidents;*Salmonella* growth dynamics during primary enrichment to improve detection and enumeration methods, with an emphasis on reducing the time to result.

## 9. Conclusions

The issue of *Salmonella* in tahini and tahini-based products is one of global concern. The tahini production continuum from sesame seed to consumption is one that transcends multiple borders, and requires international cooperation for effective management, trace-back, and source attribution. Several features of the *Salmonella*–tahini relationship make it an intractable food–pathogen combination as demonstrated by the spate of recent food safety incidents associated with it. The long shelf life and the ready-to-eat nature of tahini make it an ideal vehicle for a long and sustained exposure to *Salmonella*. Further, the high fat, low-moisture environment of tahini improves *Salmonella* survival and resiliency to a variety of lethality treatments. Given the low infectious dose of *Salmonella*, complete elimination from a batch of contaminated tahini would be required to prevent illness, a goal that is difficult to achieve in this food matrix.

The difficulty in eliminating *Salmonella* to safe levels in tahini underscores the importance of preventing contamination. For many tahini producers, roasting is the only lethal step applied in the manufacturing process. A validated roasting procedure with strict ingredient and parameter control is key. Similarly, given the high potential for contamination after roasting, process verification and environmental monitoring of the production facility is important. Correlations between total aerobic mesophile/*Enterobacteriaceae*/coliform/*E*. *coli* counts and *Salmonella* presence have not been observed, but enumeration of the former groups can provide information on the hygiene of the production process. Verification and monitoring of the entire production continuum, from sesame seed production to tahini distribution, may not be possible given the reliance on imported products and the complexity of supply chains. This complexity also complicates food safety investigations and root cause analyses in the event of food safety incidents. Therefore, it is important for the implicated countries to adhere to a set of internationally agreed-upon standards and procedures to ensure the safe manufacture and consumption of this global food product.

## Figures and Tables

**Table 2 microorganisms-10-02299-t002:** Prevalence of *Salmonella* in sesame seeds, tahini, and halva.

Product	Location	Year	Number of Positives	Total Samples	Prevalence	Reference
Sesame seed	Germany	2003	2	16	12.5%	[13]
Sesame seed	UK	2007–2008	13	771	1.7%	[59]
Sesame seed	USA	2006–2009	20	177	11.3%	[53]
Sesame seed	USA	2010	23	233	9.9	[44]
Sesame seed	USA	2013–2014	0	526	0	[52]
Sesame seed	USA	2012–2015 ^b^	12	155	7.7	[52]
Sesame seed	Mexico	2018–2019	12	100	12%	[48]
Sesame seed	Burkina Faso	2007–2017	95	359	26.5%	[49]
Sesame seed	Burkina Faso	2021 ^a^	0	25	0	[58]
Sesame seed	Sicily	2021 ^a^	36	3	8.3	[51]
Sesame seed	Portugal	2022 ^a^	0	18	0%	[57]
Tahini	Saudi Arabia	1982–1983	2 ^c^	10 ^c^	20%	[60]
Tahini	Germany	2003	1	12	8.3%	[13]
Tahini	Lebanon	2015–2017	7	42	17%	[61]
Tahini	Canada	2010–2014	9	2315	0.4%	[62,63,64]
Halva	Germany	2003	8	71	11.3%	[13]
Halva	Turkey	2010–2015	2	204 ^d^	1%	[65]
Halva	Turkey	2007–2008	0	120	0	[66]
Halva	Turkey	2004 ^a^	0	68 ^d^	0	[67]
Halva	Greece	1997 ^a^	0	4 ^c^	0	[68]

^a^ year of publication, year of sampling is not known; ^b^ collected at import; ^c^ number of production plants sampled from; ^d^ 10 g analytical unit.

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
