# Peer review of "Salmonella enterica* Outbreaks Linked to the Consumption of Tahini and Tahini-Based Products"

_microorganisms, 2022, doi:10.3390/microorganisms10112299_

Round 1

Reviewer 1 Report

The study entitled „Salmonella enterica outbreaks linked to the consumption of tahini and tahini-based products“ is a well-written article, with a lot of useful information. However, there are some gaps, which could be supplemented with additional facts. In section 2 starting from line 69, the given subtitle ’Salmonella enterica’ did not reflect the described text in terms of genus characterization, recognition of the important serovars, followed by its molecular basis, which is of particular importance for epidemiological studies, as well as for easier understanding for the readers.  In line 61 it was mentioned that this review offers a Canadian perspective on the issue of Salmonella in tahini and tahini-based products, which was substantiated with prevalence information and only mentions the whole-genome sequencing on the confirmed laboratory cases. Is there any information in terms of serotype recognition, genetic diversity, and phylogenetic analysis, which could complete the concept of the outbreak as well the traceability of the pathogens from the food sources, through the importing stage, to the final consumers? In lines 486 and  439, it was stated that a large number of samples is needed for reliable detection and prevalence determination, followed by batch-to-batch variations. In epidemiology, reliable detection is based on the determination of sample size at the expected prevalence of 50% or reduced if there is some prevalence information, along with the batch sample size determination. The variation within the batches is not the result of the sample size, but rather the influence of risk/protective factors with the initial contamination level.  The conclusion could be improved, with the marking of the risk factors, and tracking the pathogen pathway with the molecular investigation for the full characterization of the outbreak and its further incorporation in the legislation.     

Author Response

The study entitled „Salmonella enterica outbreaks linked to the consumption of tahini and tahini-based products“ is a well-written article, with a lot of useful information. However, there are some gaps, which could be supplemented with additional facts. In section 2 starting from line 69, the given subtitle ’Salmonella enterica’ did not reflect the described text in terms of genus characterization, recognition of the important serovars, followed by its molecular basis, which is of particular importance for epidemiological studies, as well as for easier understanding for the readers. 

-ST:    Thank-you for this comment. We changed subtitle to Salmonella to better reflect the text, which is intended to provide background for the reader to understand the Salmonella biology discussed in later sections of the review. We modified the text to incorporate the serovar concept. From our perspective all of the serovars are important since all members of the genus are considered pathogens. Specific serovars linked to sesame and tahini are mentioned in later sections as appropriate.

In line 61 it was mentioned that this review offers a Canadian perspective on the issue of Salmonella in tahini and tahini-based products, which was substantiated with prevalence information and only mentions the whole-genome sequencing on the confirmed laboratory cases. Is there any information in terms of serotype recognition, genetic diversity, and phylogenetic analysis, which could complete the concept of the outbreak as well the traceability of the pathogens from the food sources, through the importing stage, to the final consumers?

- ST:   These are all important and timely questions. However, this information is not available at this time, and has been identified as a data gap that requires more research in section 8.

-           In lines 486 and  439, it was stated that a large number of samples is needed for reliable detection and prevalence determination, followed by batch-to-batch variations. In epidemiology, reliable detection is based on the determination of sample size at the expected prevalence of 50% or reduced if there is some prevalence information, along with the batch sample size determination. The variation within the batches is not the result of the sample size, but rather the influence of risk/protective factors with the initial contamination level. 

-   ST:        We agree that the variation could arise from a combination of factors and having larger sample sizes may be one way to reduce the variability seen in testing: We amended the sentence as follows: These studies also demonstrated a high degree of batch-to-batch variation with respect to Salmonella presence indicating the pathogen is not distributed homogenously within the product, and may require larger sample sizes for reliable detection.  

-           The conclusion could be improved, with the marking of the risk factors, and tracking the pathogen pathway with the molecular investigation for the full characterization of the outbreak and its further incorporation in the legislation.    

ST:         Thank-you for this suggestion, we have incorporated it in the sentence: The tahini production continuum from sesame seed to consumption is one that transcends multiple borders, and requires international cooperation for effective management, trace-back, and source attribution.

Reviewer 2 Report

Dear Authors,

The review paper with the title of “Salmonella enterica outbreaks linked to the consumption of tahini and tahini-based products” reviews the Canadian perspective on the issue of Salmonella in tahini and tahini-based products and provides an overview of the available literature concerning the ecology of Salmonella in sesame seeds, tahini, and halva. I believe that this review will contribute to a deeper understanding of these criteria. As tahini and tahini-based products are primarily imported products in Canada and other immigrant states, shelf life and contamination are obvious considerations. Thanks to the authors regarding this good quality work. The paper is well organized and very interesting topic. 

According to my perspective, there is only one factor that can be explained in greater detail in the review paper, which is packaging material, since salmonella contamination occurs prior to packaging. For instance, the antimicrobial packaging materials could be one area for further research.

Other than this tiny suggestion, the paper is suitable for publication.

Good luck

Author Response

According to my perspective, there is only one factor that can be explained in greater detail in the review paper, which is packaging material, since salmonella contamination occurs prior to packaging. For instance, the antimicrobial packaging materials could be one area for further research.

Other than this tiny suggestion, the paper is suitable for publication.

ST: Thank-you for pointing this out. A sentence has been added as follows: Antimicrobial substances, such as metals, chemicals, essential oils, enzymes, and bacteriocins, can also be used in packaging materials [96]. More research is needed to evaluate their potential effect on Salmonella spp. in tahini and tahini-based products.